# A Sensing Mechanism and the Application of a Surface-Bonded FBG Dynamometry Bolt



Minfu Liang [1,2,3], Xinqiu Fang [1,2,*], Ningning Chen [1,2,*], Xiaomei Xue [4] and Gang Wu [1,2]

1   School of Mines, China University of Mining and Technology, Xuzhou 221116, China; liangmf2014@cumt.edu.cn (M.L.); wu007gang007@163.com (G.W.)
2   Research Center of Intelligent Mining, China University of Mining and Technology, Xuzhou 221116, China
3   School of Economics and Management, China University of Mining and Technology, Xuzhou 221116, China
4   School of Information and Control Engineering, China University of Mining and Technology, Xuzhou 221116, China; lb21060012@cumt.edu.cn
*   Correspondence: fangxinqiu@cumt.edu.cn (X.F.); ts15020009a3tm@cumt.edu.cn (N.C.)

**Abstract:** In the present paper, a new type of surface-bonded fiber Bragg grating (FBG) dynamometry bolt is designed. It is assumed that the adhesive layer is a linear viscoelastic material and its creep mechanical behavior is expressed by the standard linear solid model. The shear strain transfer model of the surface-bonded FBG sensor is established. Additionally, the instantaneous and quasistatic strain transfer functions of the surface-bonded FBG sensor are obtained. The functions are validated by a uniaxial tensile test and a long-term constant-load tensile test. The test results show that the strain measured by the FBG sensor has a proportional relationship with the strain measured by the resistance strain gauge. Furthermore, under the fixed load for a long period of time, the strain of the FBG sensor has a tendency to drift and the strain reduction rate is about 40.5%. Finally, the field application is carried out in a mining area. It has been proved that the ground pressure online monitoring system based on the FBG sensing technology can successfully monitor the stress of the rock bolt.

**Keywords:** fiber Bragg grating; dynamometry bolt; strain transfer efficiency; pressure online monitoring system

## 1. Introduction

Coal mining methods include the underground mining method and open-pit mining method. More than 90% of coal resources in China are obtained from underground mining. After a long period of large-scale coal mining, shallow resources are increasingly exhausted. The coal mining depth continues to extend at the rate of 10–25 m per year. The average mining depth is about 700 m and the deepest mine exceeds 1500 m [1]. The deep roadway has the characteristics of a fast deformation rate, large deformation and serious bottom drum, which significantly increases the maintenance cost of the roadway and puts forward higher requirements for the support technology. From the perspective of engineering practice, the strata behavior characteristics of roadways in different mines and different regions of the same mine are poorly similar, which is difficult to form guiding support technology. Therefore, it is necessary to construct a surrounding rock pressure monitoring system, which can collect the pressure data in real time to define a reasonable and economic support scheme.

Bolt support is a widely used active support method, which contributes to great economic benefits for Chinese coal enterprises [2]. With the wide application of bolt support, many scholars are devoted to the study of its mechanical characteristics and supporting mechanism through different detection methods. In reference to [3–5], the stress wave non-destructive detection method, electromagnetic method, and infrared radiation detection method were used to detect the quality of rock bolts. Nowadays, it is widely

recognized to monitor the axial force of the anchor rod by the dynamometry bolt. According to the different measurement principles, the conventional force-measuring anchors could be divided into two types: the electric measuring type and mechanical measuring type. The force measuring bolts equipped with a resistance strain gauge were one type of the electric measuring type, which were also the most common in engineering practice. Ding et al. studied the stress distribution law and supporting mechanism of the force measuring bolts in the process of rock mass deformation. The force measuring bolts were installed manually and instrumented five pairs of symmetrical strain gauges [6]. The above detection methods had the following drawbacks: they were susceptible to external interference, complicated signal processing, unable to achieve real-time monitoring, unable to achieve distributed measurements, and unfavorable to long-distance information transmission. Hence, this paper aims to study a new type of force-measuring bolt, and construct a surrounding rock pressure monitoring system. Coal mine technicians only need to check the axial force of the bolt in the ground command center, and there is no need to arrange special personnel to collect the data on site.

In the early 1970s, the optical fiber sensing (OFS) technology that used light waves as carriers for sensing and measurements came into being. The OFS technology was suitable for real-time online detection and has been applied to the underground engineering of coal mines, which initially realized the measurement of rock settlement and underground excavations, support loads, surrounding rock mass stress, and disaster prediction [7–10]. Some scholars carried out research on the detection of the bolt body. Chai et al. [11] studied the performance of the bolt under pull-out loading using the pulse pre-pump Brillouin optical time domain analysis (PPP-BOTDA) and fiber Bragg grating (FBG) sensing technologies. Wang et al. [12] designed a full rod FBG force-measuring bolt and system based on the principle of fiber grating sensing to continuously monitor the axial force of the bolt over a long period of time. However, Chai and Wang neglected the influence of binders on the sensing characteristics of the FBG force-measuring bolt. Duck et al. [13] presented a theoretical model for the shear strain transfer of the embedded FBG sensors, based on the assumption that the strain of the host material was the same as the strain of the center of the FBG, but they ignored the stress coupling effect of the adhesive layer. Li et al. [14] considered the viscoelasticity of the adhesive layer and proposed a time-dependent equation of strain transfer. Their research results also had an important impact on this paper.

In this paper, a new type of FBG dynamometry bolt is designed. The difference with the FBG sensor designed by predecessors is that the FBG sensor studied in this paper adds a desensitization device. Taking the ground pressure monitoring of return airway 81303 in the Yangquan No. 1 coal mine as the engineering background, we establish the FBG online monitoring system for the ground pressure, and focus on the transmission mechanism and sensing characteristics of FBG dynamometry bolt. Furthermore, the theoretical analysis previously mentioned is validated by a uniaxial tensile test and a long-term constant load tensile test. Finally, it is confirmed that the FBG dynamometry bolt can replace the traditional resistance dynamometry bolt to monitor the stress of the bolt body by four characteristic indexes: the correlation coefficient, measuring range, linearity, and sensitivity.

## 2. The Sensing Theory of FBG Dynamometry Bolt

### 2.1. Optical Fiber Sensing Principle

In this paper, the sensing transmission mechanism of the FBG sensor bonded on the surface of the matrix structure via adhesives is studied. According to the shear lag theory proposed in the literature [13] to analyze the internal stress transfer of composites, it was considered that the strain was transmitted from the substrate to the fiber grating in the form of shear strain through the adhesive layer. On the one hand, the FBG sensor packaging materials are mostly polymers. The magnitude order difference between the elastic modulus of polymers and the elastic modulus of the optical fiber and the substrate results in a non-synergistic deformation between the response strain and the actual strain. On

the other hand, most polymers are viscoelastic, which can show the mechanical properties of elastic solids and viscous liquids under the response of an external force, that is, their mechanical properties change over time. The study of the strain transfer rules of the FBG sensors over time can provide practical and reliable theoretical analyses for parameter calibration, accuracy analysis, error estimation, and correction of sensors, so that the sensors can maintain good performance for long-term service in engineering practice.

A strong ultraviolet laser was used to change the structure of the germanium-doped fiber core, which can form fringes with periodic changes in the refractive index and are uniformly distributed along the axial direction inside the fiber; a diffraction grating is then formed. The FBG sensors are composed of the fiber core, cladding layer, coating layer, reinforced fiber layer, and optical fiber protective sleeve from the inside to the outside. The fiber without a reinforced fiber layer and fiber protective sleeve is called bare fiber optics. The structure is shown in Figure 1. The main component of the fiber core is silica, which is doped with a very small amount of germanium. The main component of the cladding layer is pure silica. The coating layer is generally a polymer material, such as epoxy resin or silicone rubber.

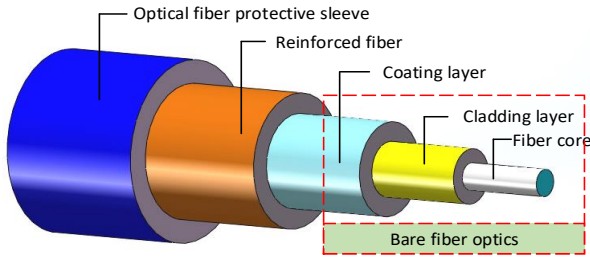

**Figure 1.** Schematic diagram of the FBG sensor's structure.

As shown in Figure 2, when a wide spectrum incident light passes through the fiber grating, the light waves meeting the Bragg condition of the fiber grating are reflected, and the rest of the light waves continue to be transmitted through the optical fiber. The reflected light is demodulated by the optical element and the peak of the reflected light wavelength is called the center wavelength $\lambda_B$. When the external physical quantity changes, the grating period $\Lambda$ and effective refractive index $\Delta n_{\text{eff}}$ of the grating part are changed simultaneously, and then the reflected wavelength changes accordingly. The change of the reflected wavelength and the external physical quantity follow a fixed law, so that the value of the external physical quantity can be reflected by the change of the central wavelength.

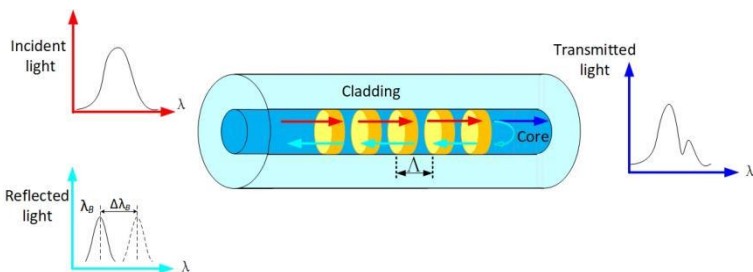

**Figure 2.** Principle of fiber Bragg grating sensing.

When the optical fiber is only subjected to stress, the change of the central wavelength of the reflected light has the following relationship with the axial strain of the grating [15]:

$$\frac{\Delta\lambda_{\text{B}}}{\lambda_{\text{B}}} = (1 - P_{\text{e}})\varepsilon, \tag{1}$$

where $\Delta\lambda_B$ is the drift value of the central wavelength of the reflected light, $\varepsilon$ is the axial strain of the grating, and $P_e$ is the elasto-optical coefficient. For silica fiber, $P_e$ is equal to 0.22.

When the optical fiber is only affected by temperature, the change of the central wavelength of the reflected light has the following relationship with the temperature:

$$\frac{\Delta\lambda_B}{\lambda_B} = (\alpha + \zeta)\Delta T, \tag{2}$$

where $\alpha$ is the thermal expansion coefficient, $\zeta$ is the thermo-optic coefficient, and $\Delta T$ is the temperature variation.

When the optical fiber is under the combined action of stress and temperature, the change of the central wavelength of the reflected light has the following relationship with the independent variable:

$$\frac{\Delta\lambda_B}{\lambda_B} = (1 - P_e)\varepsilon + (\alpha + \zeta)\Delta T, \tag{3}$$

### 2.2. Design of FBG Dynamometry Bolt

#### 2.2.1. Design Principles

The rock bolt is generally made of high-strength steel, which can withstand high stress loads. Its elongation is in the range of 15–25%. The elongation of the test bolt in the present study is about 20%, measured by a pre-tensile test. The optical fiber is a brittle material and its elongation is only 0.3%. If the fiber Bragg grating is directly adhered to the bolt surface, it can be easily broken. Therefore, it is necessary to add desensitization devices to protect the optical fiber, which can perceive a wide range of deformation.

#### 2.2.2. The Structure of the FBG Dynamometry Bolt

The simple structure and principle of the spring make it ideal for use as a desensitization device. In order to firmly fix the spring to the bolt rod, we decided to cut a rectangular groove with a width of 4 mm and a depth of 4 mm along the length direction of the bolt rod and then leave 6 limit slots in the rectangular groove. Each spring under little tension was positioned in the corresponding limit slot. Both ends of the spring were connected with the bolt rod by epoxy glue or welding. An optical fiber engraved with six sections of grating (1# to 6#) was customized. The grating part was glued to the middle of the spring, correspondingly, and the length of the adhesive was 12 mm. The end of the optical fiber was connected to the tail fiber protected with armored rubber and led out from the end of the bolt. Finally, the rectangular groove was sealed with epoxy glue. In order to ensure the long-term and effective monitoring of the FBG dynamometry bolt, we needed to polish the groove so that it was smooth, to ensure that the groove was flat and clean. Before sticking the fiber grating, the groove and the optical fiber should be cleaned with acetone or alcohol. Figure 3 shows the structure diagram of the limit slot for the FBG dynamometry bolt, and Figure 4 shows the schematic diagram of the fiber grating arrangement on the bolt body.

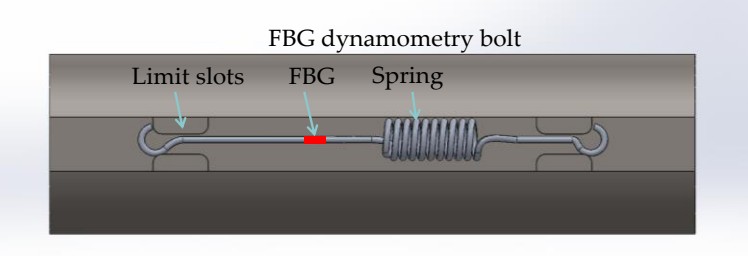

**Figure 3.** Structure diagram of the limit slot for the FBG dynamometry bolt.

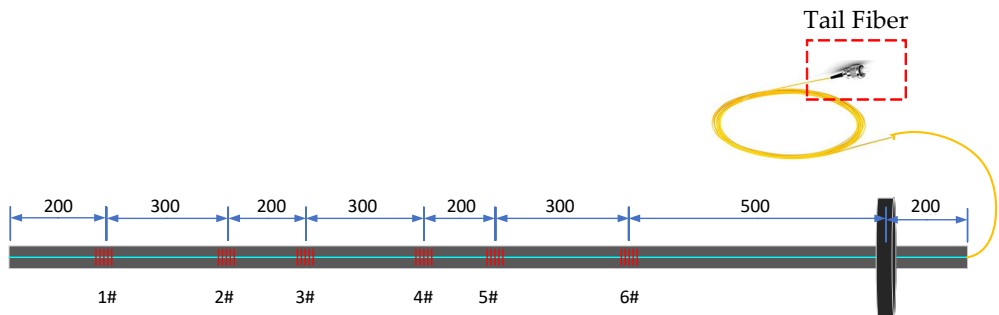

**Figure 4.** Schematic diagram of the fiber grating arrangement on the bolt body.

### 2.3. The Working Principle of the FBG Dynamometry Bolt

The bolt and the spring were simultaneously deformed, and the spring was stretched to generate stress. The spring stress was transferred to the fiber core in the form of shear stress through the adhesive. After the fiber core was stressed, the grating period and effective refractive index of the grating part were changed. The reflected optical signal carrying the bolt strain information was transmitted to the demodulator host through communication equipment, such as jumpers, optical splitters, and multi-core transmission optical cables. The optical signal was demodulated into electrical signal data and transmitted to the ground server for storage through the underground ring network. The staff accessed the data through the client of the optical fiber monitoring system to realize the real-time online monitoring of the force of the bolt body. The working principle of the FBG dynamometry bolt is shown in Figure 5.

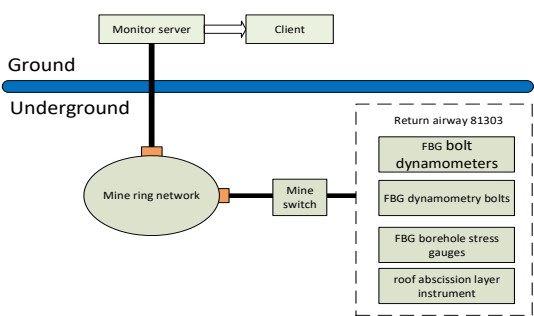

**Figure 5.** Data transmission schematic diagram of the FBG dynamometry bolt.

The bolt and the spring deform together, and the surface strain of the bolt is equal to the spring strain, and the relationship is as follows:

$$\varepsilon_{\mathrm{m}} = \varepsilon_{\mathrm{s}} = \frac{L_{\mathrm{s}}}{L_{\mathrm{f}}} \varepsilon_{\mathrm{h}}, \tag{4}$$

where $\varepsilon_{\mathrm{m}}$, $\varepsilon_{\mathrm{s}}$, $\varepsilon_{\mathrm{h}}$, respectively, represent the strain of the bolt, the total strain of the spring, and the strain of the spring matrix of the section that overlaps with the fiber grating. $L_{\mathrm{s}}$, $L_{\mathrm{f}}$, respectively, indicate the full length of spring, when it is completely straightened, and the paste length of the fiber grating.

The spring matrix strain cannot be completely transmitted to the fiber core and there is a loss in the process of transmitting. It is necessary to study the strain transfer efficiency. The efficiency value of the spring matrix strain transferring to the fiber Bragg grating is expressed by the following mathematical formula:

$$\eta = \frac{\varepsilon_{\mathrm{c}}}{\varepsilon_{\mathrm{h}}}, \tag{5}$$

where $\eta$ represents the efficiency value of the spring matrix strain transferring to the fiber Bragg grating; $\varepsilon_c$ represents the strain sensed by the fiber Bragg grating.

### 3. Strain Transfer Efficiency of the Surface-Bonded FBG Sensor

*3.1. Strain Analysis Model of the FBG Sensor*

The schematic diagram of the FBG sensor structure is shown in Figure 6, where D, H, and h represent the width of the adhesive layer, the thickness of the adhesive layer, and the thickness of the adhesive interlayer, respectively. $r_c$ and $r_p$ are the radii of the bare fiber and the cladding layer.

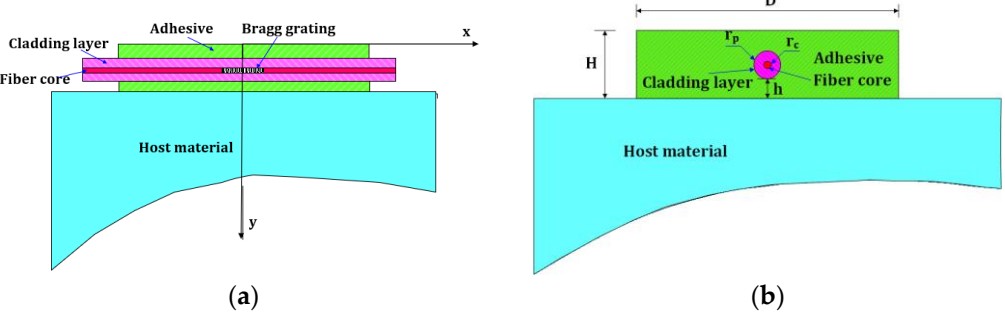

|        |        |
|:------:|:------:|
| **(a)** | **(b)** |

**Figure 6.** Schematic diagram of the sensor structure. (**a**) The main view of the schematic diagram; (**b**) the side view of the schematic diagram.

After a mechanical analysis and formula derivation, the strain transfer efficiency ($\eta$) of the FBG strain transfer and error correction factor ($k$) are defined as

$$\eta = \frac{\overline{\varepsilon_c(x,t)}}{\varepsilon_0} = L^{-1}\left[\frac{1}{s}\left(1 - \frac{2\tanh(\overline{\xi}L/2)}{\overline{\xi}L}\right)\right],\tag{6}$$

$$k = \frac{1}{1-\eta},\tag{7}$$

$$\text{where}\, \frac{1}{\overline{\xi}^2} = s\cdot\overline{J}_a(s)\cdot\frac{(2\pi r_p + D)r_p E_c h}{4D} + \frac{(r_p + r_c)r_p E_c}{4G_p},\tag{8}$$

*3.2. Analysis of the Mechanical Model of the Adhesive Layer*

In order to describe the viscoelastic mechanical behavior of the adhesive layer more accurately and study the sensing response characteristics of the FBG sensors under long-term fixed loads, we assume that the adhesive layer is a standard linear solid model consisting of a Kelvin model and a spring, as shown in Figure 7, where $E_{a1}$ and $E_{a2}$ represent the elastic modulus of the spring element and the elastic modulus of the Kelvin element, respectively. $\eta_2$ represents the viscosity coefficient of the Kelvin element.

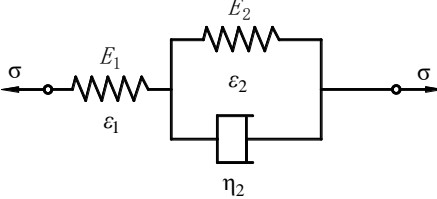

**Figure 7.** A standard linear solid model.

According to the literature [16], we can obtain

$$\xi_0 = \left(\frac{1}{2G_{a1}} \cdot \frac{(2\pi r_p + D)r_p E_c h}{4D} + \frac{(r_p + r_c)r_p E_c}{4G_p}\right)^{-\frac{1}{2}}, \tag{9}$$

$$\xi_\infty = \left(\frac{G_{a1} + G_{a2}}{2G_{a1}G_{a2}} \cdot \frac{(2\pi r_p + D)r_p E_c h}{4D} + \frac{(r_p + r_c)r_p E_c}{4G_p}\right)^{-\frac{1}{2}}, \tag{10}$$

The instantaneous and quasi-static strain transfer efficiency of the FBG sensor are, respectively:

$$\eta_0 = \frac{\overline{\varepsilon_c(x,0)}}{\varepsilon_0} = 1 - \frac{2\tanh(\xi_0 L/2)}{\xi_0 L}, \tag{11}$$

$$\eta_\infty = \frac{\overline{\varepsilon_c(x,\infty)}}{\varepsilon_0} = 1 - \frac{2\tanh(\xi_\infty L/2)}{\xi_\infty L}, \tag{12}$$

Finally, the bolt strain under the instantaneous and quasi-static states can be obtained as

$$\varepsilon_m \approx \frac{\Delta\lambda_B L_s}{\eta\lambda_B(1-P_e)L_f}, \tag{13}$$

where $\eta$ is equal to $\eta_0$ in the instantaneous state and $\eta$ is equal to $\eta_\infty$ in the quasi-static state.

For the specific derivation process of Equations (6)–(12), please refer to the published literature [17].

## 4. Characteristic Test and Analysis of the FBG Dynamometry Bolt

### 4.1. Tensile Test of the Dynamometry Bolt

In order to verify the correctness of the above formula (13), two groups of bolt tensile tests were designed. The electro-hydraulic servo testing machine is used to perform tensile loading on the bolt sample. The size of the FBG dynamometry bolt sample is Φ20 mm×840 mm. In test I, two measuring positions were arranged on the bolt body. The 2 measuring positions were 200 mm apart, and each of them was 320 mm away from the end of the bolt sample, respectively. In test II, only one measuring position was arranged and it was located in the middle of the bolt sample. A resistance strain gauge sensor and a FBG sensor were pasted at each measuring position. Using the strain measured by the strain gauge as a reference, the center wavelength shift ($\Delta\lambda_B$) of the reflected light measured by the FBG sensor was substituted into formula (13) to obtain the measured strain. Additionally, the measured strain of the resistance strain gauge sensor and the measured strain of the FBG sensor were compared and analyzed to study the rule of the sensing characteristics for the FBG dynamometry bolt. In addition, the adhesive creeps under the long-term loads, which affects the stress state of the fiber grating, so it is very important to study the stability of the FBG dynamometry bolt under the long-term tensile loads.

The diagram of the test device and the schematic diagram of the connection of the test equipment are shown in Figures 8 and 9.

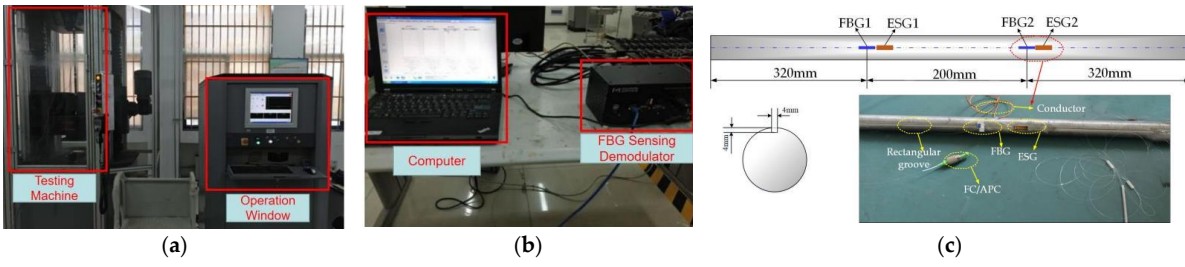

**(a)**      **(b)**      **(c)**

**Figure 8.** The diagram of the test device. (**a**) Electro-hydraulic servo testing machine; (**b**) FBG sensing demodulator and static resistance strain demodulator; and (**c**) the sample of the FBG dynamometry bolt.

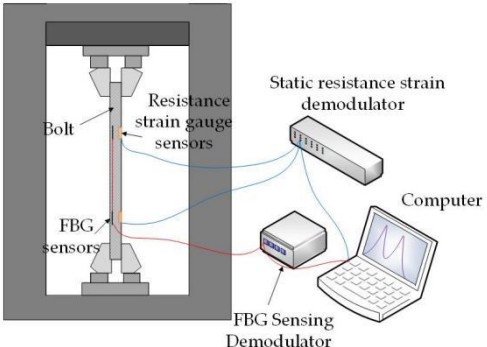

**Figure 9.** Schematic diagram of the test equipment connection.

*4.2. Test Scheme*

The test scheme included the following five steps:

1.  Connected and debugged the instrument: the FBG static demodulator was connected to the computer through the network cable and completed the debugging on the demodulation software, and the data storage path was set. The resistance strain gauge was connected to the computer through an RS485 bus, zeroed, and calibrated, and the data storage path was set at the same time.
2.  Set the loading methods: in test I, the loading test machine was set to load at the rate of 5 mm/min, and maintained this for 2 min when it was loaded to 5 kN, and then it continued to perform the loading method. At this time, the measurement data of the resistance strain gauge and the FBG sensor were recorded and stored until the data could not be collected on the FBG sensing demodulator and resistance strain gauge. In test II, the loads were, respectively, loaded to 120 kN, 140 kN, and 160 kN within 1 min (the tensile force of the bolt sample when it yielded was measured to be 170 kN during the pre-test), and the load was kept unchanged. After maintaining for 2 h, the load was then unloaded to 5 kN within 1 min.
3.  The bolt sample was clamped and prepared for the test: first, the upper clamp of the testing machine was opened, the sample was placed into the clamp groove, and then the upper clamp was closed to clamp the sample. Then, the lower clamp of the testing machine was opened, the lower clamp was raised to the end of the sample through the remote control switch, and then the lower clamp was closed to clamp the sample.
4.  Begin the test: the testing machine, the FBG demodulator, and the resistance strain gauge demodulator recorded the measurement data at the same time.
5.  The data was exported from the computer and saved.

*4.3. Test Results and Discussion*

4.3.1. Analysis of the Results and Discussion of Test I

The entire test process lasted for 20 min. The variation curve of the measurement data sensed by the testing machine, two groups of resistance strain gauges, and two groups of FBG sensors over time are shown in Figure 10.

It can be seen in Figure 10 that the tension bolt in this test has experienced the elastic deformation stage, the yield deformation stage, and the plastic deformation stage. The tensile force when the test bolt yields is about 175 kN. In the elastic stage, the strain changes linearly with time. In the yield stage, the strain is evidently stable within a certain range. In the plastic stage, the strain slowly increases with time, and the deformation rate gradually decreases.

The resistance strain gauge sensor can monitor the linear elastic process and the entire yield process of the tension bolt. After yielding, the strain gauge sensor fails. The failure of the strain gauge is due to the failure of the adhesive, which leads to the separation of the strain gauge from the anchor rod. However, due to the small size of the optical fiber, the adhesive can closely connect the optical fiber with the spring. Additionally, the spring

can also protect the optical fiber, so that the FBG dynamometry bolt can monitor the linear elastic stage, the yield stage, and the partial plastic stage of the bolt tensile process. In the linear elastic stage, the sensing curve is almost straight. In the yield stage, the perceived strain is roughly maintained within a certain range and a jumping "wavelet peak" appears at the end of each yield. The "wavelet peak" can be used as a warning sign to detect the end-of-the-yield stage of the bolt rod. Then, at a certain time of plastic deformation, the adhesive between the spring and the bolt rod fails, which results in the failure of the FBG dynamometry bolt.

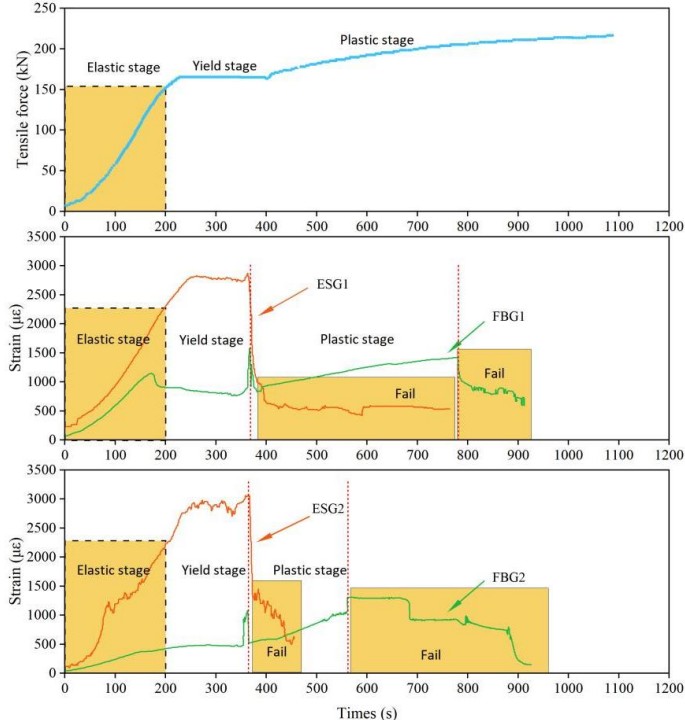

**Figure 10.** Measurement results of each test instrument.

After substituting the central wavelength measured by the FBG sensor in the elastic stage into Equation (13), the calculated strain of the FBG dynamometry bolt is the dependent variable, and the strain measured by the resistance strain gauge sensor in the elastic stage is the independent variable. The relationship curve between the dependent and independent variable is shown in Figure 11. The physical and mechanical parameters of the experimental materials are shown in Table 1.

**Table 1.** The physical and mechanical parameters of the experimental materials.

| Material Parameters | Physical Symbols | Values | Units |
|---|---|---|---|
| Radius of bare fiber core | $r_c$ | 62.5 | μm |
| Radius of the cladding layer | $r_P$ | 125 | μm |
| Elastic modulus of the fiber | $E_c$ | 72 | GPa |
| Shear modulus of the cladding layer | $G_P$ | 400 | MPa |
| Bond length | $L$ | 12 | mm |
| Spring length | $L_s$ | 60 | mm |
| Bond width | $D$ | 2 | mm |
| Thickness of the adhesive interlayer | $h$ | 0.2 | mm |
| Shear creeps compliance | $G_{a1}$ | 5 | GPa |
| Central wavelength initial value | $\lambda_{B1}, \lambda_{B2}$ | 1515.694 1531.054 | nm |

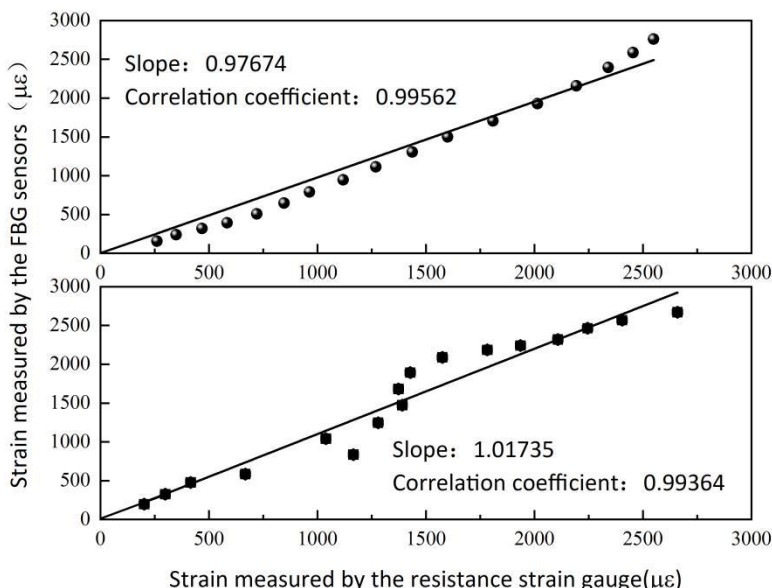

**Figure 11.** Corresponding relationship of the strain measured by the resistance strain gauge sensors and FBG sensors.

The slopes of the 2 straight lines obtained by fitting are 0.97674 and 1.01735, which are approximately equal to 1. It shows that the strain measured by the FBG dynamometry bolt and the resistance strain gauge have a proportional relationship. At the same time, it can also be explained that the perceived strain calculated by Equation (13) is approximately equal to the theoretical value. The feasibility of Equation (13) is demonstrated.

The correlation coefficient, range, linearity, and sensitivity of the resistive strain gauge and the FBG sensor are presented in Table 2.

**Table 2.** A comparison of the indicators in the three groups of tests.

|  | Resistance Strain Gauge 1 | FBG Sensor 1 | Resistance Strain Gauge 2 | FBG Sensor 2 |
|---|---|---|---|---|
| Range (/kN) | 166 | 166 | 167 | 167 |
| Correlation coefficient | 0.99814 | 0.99617 | 0.99347 | 0.99553 |
| Linearity | 10.24% | 6.84% | 18.68% | 11.0% |
| Sensitivity ($\mu\varepsilon$/kN) | 16.44 | 16.09 | 16.73 | 18.56 |

By analyzing Table 2, it can be concluded that the FBG sensor is close to the reference value of the resistance strain gauge in the above four indexes, which verifies that the designed FBG dynamometry bolt can replace the traditional resistance dynamometry bolt to monitor the stress of the bolt body. In terms of the measuring range and linearity, the FBG sensor is better than the strain gauge sensor.

4.3.2. Results Analysis and Discussion of Test II

Figure 12 shows the curve graph of the strain of the sample under the long-term tensile loads measured by the resistance strain gauge sensor and the FBG dynamometry bolt in Tests II-1, II-2, and II-3.

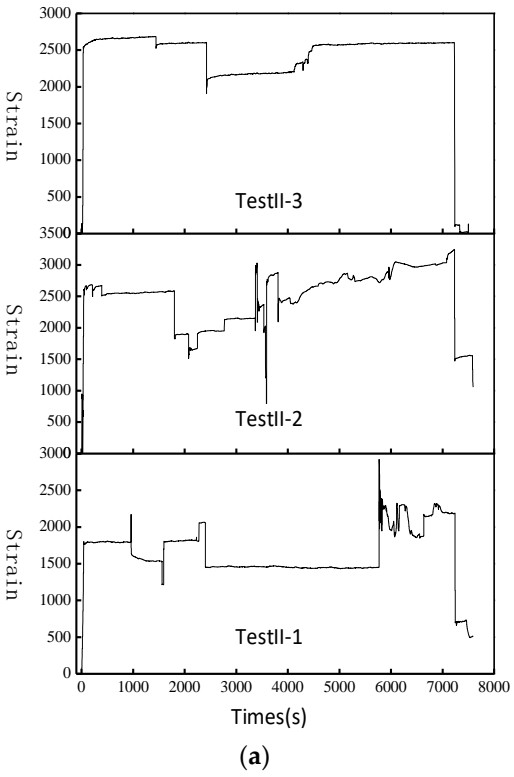

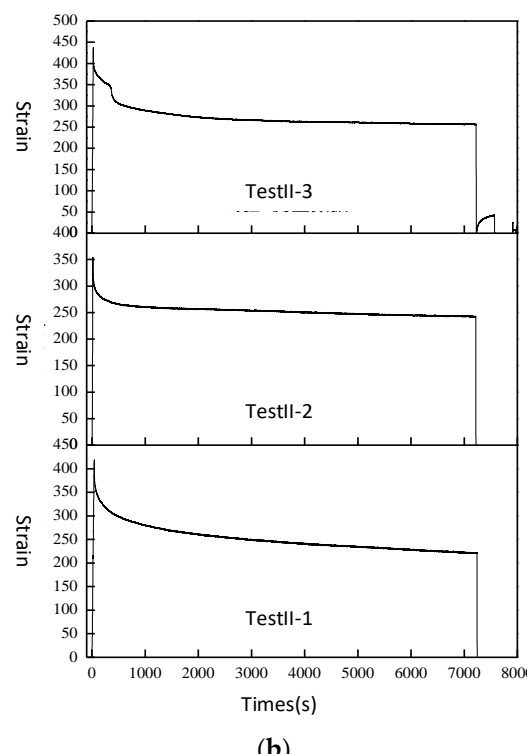

(a)                                                                (b)

**Figure 12.** Variation curve of the specimen strain with time under the long-time constant loads. (**a**) The measurement results of the resistance strain gauge sensors; (**b**) the measurement results of the FBG sensors.

It can be observed in Figure 12 that, under the action of fixed tensile loads over a long period of time, the strain values of the strain gauge sensor and FBG sensor drift. From the trend change of the strain curve, the resistance strain gauge sensor is unstable and can only maintain a stable state for a short period of time. In fact, the resistance strain gauge is calibrated before use. If the axial force of the anchor bolt is still calculated according to the calibration results, it leads to an incorrect judgment. In contrast, the FBG sensor has a specific strain drift law and the stable strain can be calculated through the quasi-static equation of Equation (13), which does not misjudge the axial force of the bolt rod.

The strain curve of the FBG dynamometry bolt follows an obvious law change, which shows a nonlinear, smooth decline and gradually tends to be stable on the whole. When the tensile force reaches the predetermined value, the specimen strain reaches the maximum values of 418 $\mu\varepsilon$, 354 $\mu\varepsilon$, and 437 $\mu\varepsilon$. The strain rapidly attenuates in the first 1000 s, reaching 220 $\mu\varepsilon$, 241 $\mu\varepsilon$, and 256 $\mu\varepsilon$ at 1000 s, and then the curve becomes flat and drops to 220 $\mu\varepsilon$, 241 $\mu\varepsilon$, and 256 $\mu\varepsilon$ by 7200 s. The strain attenuation values of the FBG dynamometry bolt are 198 $\mu\varepsilon$, 113 $\mu\varepsilon$, and 184 $\mu\varepsilon$, respectively, and the attenuation rates are 47.4%, 31.9%, and 42.1%.

## 5. Engineering Application

### 5.1. Engineering Background

The No. 15 coal seam is mined in the Yangmei No. 1 Coal Mine, affiliated with the Shanxi Huayang Group New Energy Co., Ltd. (Yangquan, China), with an average burial depth of 540 m. The return airway 81303 was selected as the engineering test face, which is adjacent to the unmined working face 81305 in the north, the goaf of the working face 81301 in the south, the main roadways in the west, and the mining area boundary in the east. The 81303 return airway is designed with a rectangular section, with a net width of 5000 mm, a net height of 3700 mm, and a net cross-section of 18.5 m².

The three comprehensive measuring stations are arranged in the return airway 81303. Each measuring station is equipped with seven FBG bolt dynamometers, three FBG dynamometry bolts, four FBG borehole stress gauges, and two groups of roof abscission layer instruments. The layout of the measuring stations is shown in Figure 13.

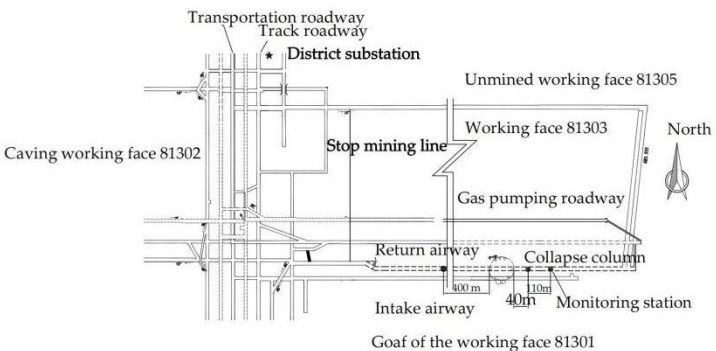

**Figure 13.** The roadway section deformation monitoring station layout in the return airway 81303.

The FBG dynamometry bolts are arranged on the roof and two sides of the roadway, and the on-site connection and installation of the dynamometry bolts are shown in Figure 14.

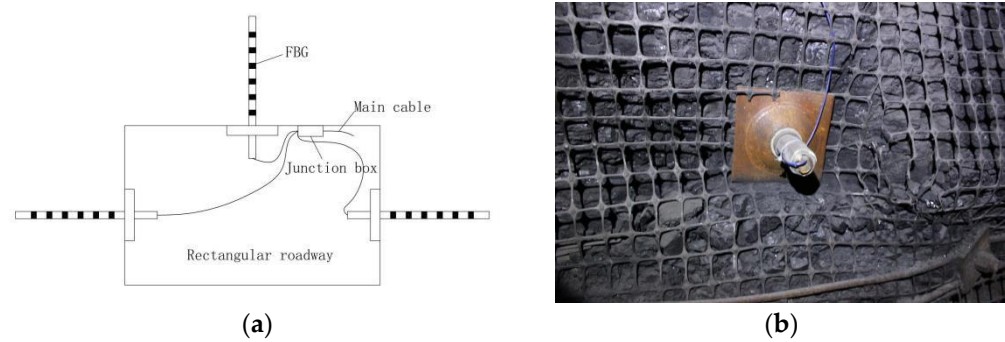

(**a**)                                                                    (**b**)

**Figure 14.** The on-site connection and installation of the dynamometry bolts. (**a**) Device connection diagram; (**b**) on-site installation drawing.

### 5.2. Monitoring Data Analysis

The stress monitoring of the bolt body is an important part of roadway support monitoring. Through the FBG online monitoring system of ground pressure, it is convenient to extract the data of each sensing equipment at each underground measuring station. An early warning value of the bolt force as preset in the FBG online monitoring system. When the bolt force reaches or exceeds this warning value, the system automatically triggers an alarm. This early warning value is set as the maximum force value in the elastic deformation stage of the bolt.

Figure 15 shows the stress monitoring data of the rock bolt body on the roadway roof, solid coal side, and narrow coal pillar side at positions 5, 20, 50, 70, 90, and 110 m from the head of the tunnel face.

From Figure 15, it can be analyzed that the roadway is in the excavation influence period, from 5–20 m away from the head, the tensile force of the roof bolts and the 2 sets of bolts are small, and the force distribution of the bolt body is relatively uniform. The stress of the roof bolt reaches the maximum at 70 m from the head and the maximum stress of the bolt occurs at 800 mm from the periphery of the roadway. The stress of the bolt for the solid coal side reaches the maximum at 90 m from the head, and the maximum stress occurs at about 500 mm from the surrounding of the roadway. The stress of the bolt for the

narrow coal pillar side reaches the maximum at 90 m from the head, and the maximum stress occurs at about 1000 mm from the periphery of the roadway.

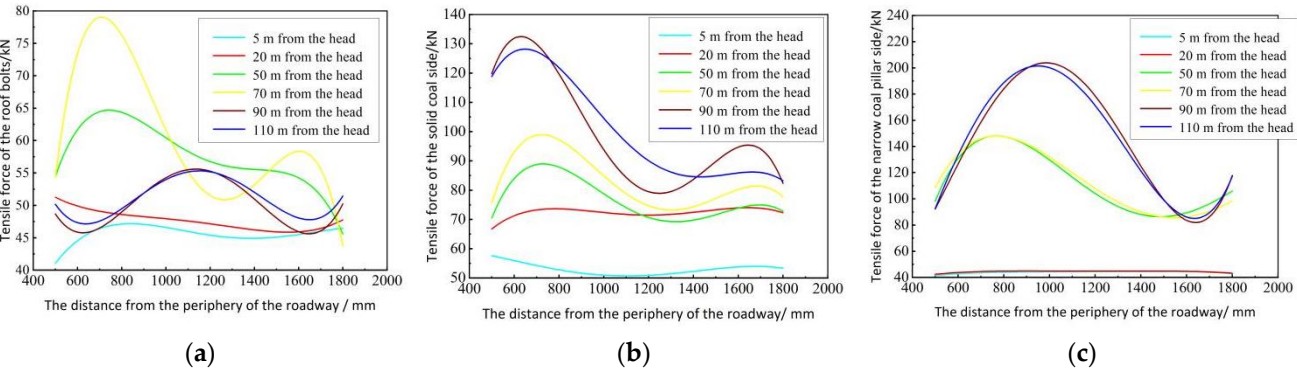

**Figure 15.** The stress of the bolt in different positions from the head of the tunnel face. (**a**) The roof bolts on the roadway, (**b**) the bolts on the solid coal side, and (**c**) the bolts on the narrow coal pillar side.

## 6. Conclusions

In this paper, it is assumed that the adhesive layer is a linear viscoelastic material and its creep mechanical behavior is expressed by the standard linear solid model. The shear strain transfer model of fiber Bragg grating is established and the suitability of optical-mechanical conversion equation of the surface-bonded FBG dynamometry bolt for coal mine safety monitoring is deduced. In order to verify the correctness of the above Equation (13), a uniaxial tensile test and a long-term constant load tensile test are designed. The results of test I show that the strain measured by the FBG dynamometry bolt and the resistance strain gauge have a proportional relationship and verify that the FBG dynamometry bolt can replace the traditional resistance dynamometry bolt to monitor the stress of the bolt body. In terms of the measuring range and linearity, the FBG sensor is better than the strain gauge sensor. In test II, it is determined that under fixed loads, for a long period f time, the strain of the FBG sensor drifts and the strain reduction rate is about 40.5%. The measurement result for the FBG sensor reflects the axial force of the anchor bolt more accurately than that of the resistance strain gauge sensor. However, the strain drift phenomenon of the FBG sensor needs to be further studied to eliminate the measurement error introduced by the strain drift. Finally, the field application is carried out in the return airway 81303 of Yangmei No. 1 Mine. It has been proved that the monitoring system can successfully monitor the stress of the rock bolt body in the underground roadway.

**Author Contributions:** Conceptualization, M.L. and X.F.; methodology, N.C.; validation, M.L., N.C., X.F., X.X. and G.W.; data curation, N.C.; writing—original draft preparation, N.C.; writing—review and editing, M.L., N.C., X.F., X.X. and G.W.; funding acquisition, M.L., X.F. and G.W. All authors have read and agreed to the published version of the manuscript.

**Funding:** This work was supported by the National Natural Science Foundation of China (Nos. 52004273, 51874276 and 52104167), the Natural Science Foundation of Jiangsu Province (No.BK20200639), the China Postdoctoral Science Foundation (No.2019M661992), and the Fundamental Research Funds for the Central Universities (No.2020QN38).

**Institutional Review Board Statement:** Not applicable.

**Informed Consent Statement:** Not applicable.

**Data Availability Statement:** The study did not report any data.

**Conflicts of Interest:** The authors declare no conflict of interest.

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
