# Peer review of "A Sensing Mechanism and the Application of a Surface-Bonded FBG Dynamometry Bolt"

_applsci, doi:10.3390/app12073225_

Round 1

Reviewer 1 Report

The manuscript describes a Fiber Bragg dynamometry bolt sensor to monitor the stress of the bolt body by four characteristic indexes of cor-74 relation coefficient, measuring range, linearity and sensitivity. It starts by providing theoretical model of the bolt design parameters followed by two lab tests in comparison to resistance strain sensor. Later an experimental application was studied on a mine roadway section in China.

The manuscript is interesting with moderate English language style. I have some comments on the article that needs the attention of the authors prior to considering the article suitable for publication:

-The introduction is short and does not include historical background and other topologies used for the problem at hand. I encourage the authors to provide more info about state of the art from other publications.

-The authors did not mention clearly the advantage of the proposed FBG method in comparison to the resistance strain method or other topologies. They just concluded that the FBG sensor can replace the resistance strain sensor, but why would the industry go for this option with the complicated design and fabrication. They need to highlight more info in this regard.

-I was expecting to see a comparison in section 5 similar to the one in Table 2 but unfortunately the authors did not use the resistance strain gauge comparison in the engineering experiment. If possible to add this experimental comparison, or at least to mention why it is not included considering they trust the FBG after the results of section 4.3

I would advise that the article needs a minor revision before considering it ready for publication.

Reviewer 2 Report

The presented work is devoted to methods of ensuring safety in mining operations, which is an urgent task of eliminating or, at least, significantly reducing the likelihood of technological accidents.

But, first of all, the article is very poorly written, both in terms of the English language and the organization of the presentation. This greatly complicates the understanding of the work content.

Secondly, the scientific novelty of the work and its main results are not clarified. The use of fiber optic sensors of FBG type in distributed strain monitoring systems has long been well known.

If the results of the work consist in testing a new strain control device on known principles, then the article does not correspond to the subject of the journal Applied Sciences.

There is a lot of information in the text that is not related to the purpose of the work.

The description "4.2 Test Scheme" resembles the instructions for the testing machine, and not like a scientific experiment.

Section "5 Engineering Application" uses a lot of highly technical terms related to mining and not understandable to a wider circle of Applied Sciences readers.

Some local remarks:

line 118

It is necessary the clarification: e – is the amount of relative axial strain (ΔL/L)

line 123

What difference in α and ζ? "Where, α is the thermal expansion coefficient and ζ is the thermal expansion coefficient also..."

line 151

The design of the FBG dynamometry bolt is not clear. The position of the fiber in Figure 3 is not indicated. What are the reasons for choosing the spring hardness?

line 153

What do the numbers in Fig. 4 mean? How do Figures 3 and 4 compare?

line 185

Where in Fig.6 rc и rp?

line 225

Figures 8(a) и 8(b) do not carry any information. Figure 8(c) is very small and not legible.

line 261

Is the link to Figure 12 correct?

line 325

What are the units of με? What System of Units do they fit?

line 343

The inscriptions in Fig. 13 are too small and cannot be read. The purpose of this figure is not clear.

and so on...

Reviewer 3 Report

This paper presents a design of a sensing mechanism and application of surface-bonded fiber bragg grating (FBG) Dynamometry Bolt. The topic is certainly worthy of investigation nowadays, but the manuscript suffers from the following shortcomings that need to be thoroughly addressed before it could be considered for publication:

  • The research question and has not been put forward clearly.
  • Both the level and context of the introduction would only benefit researchers who have just started working in this subject. Importantly, some of what is presented in not being correlated to on-going research and technology developments. Therefore, recent references should be considered.
  • A comparison table between the proposed system and other related studies is highly recommended to be added in Section 2. Then, motivations can be easily drawn.
  • The structure of FBG dynamometry bolt should be explained and justify why has been selected?
  • Overall, although the authors designed an interesting experiment, the results are poorly discussed. To this end, the experiments are not convincing.
  • The manuscript would greatly benefit from proofreading, as there are lots of grammar errors, vague statements and claims, and typos.
  • Figures could be improved from size and resolution perspectives, especially Figures: 9, 10, 13, 14 and 15.
  • There are some self-citations of the authors' works. Please revise this carefully to the most relevant papers.
  • Overall, the manuscript shows moderate work and it can be considered after major revision.

Round 2

Reviewer 2 Report

Minor corrections made in the text do not change the quality of the results presentation. The results presented in Figure 11 indicate the absence of a linear relationship between the resistance strain gauge and the FBG gauge, contrary to the conclusions of the authors. The study of sensors in the viscoelastic or inelastic deformation conditions does not make practical sense, since the sensor changes its properties irreversibly and deprives further measurements of reliability.

Round 3

Reviewer 2 Report

No significant corrections or additions are presented. The work needs a significant revision not only of the text, but also of the experiments. As presented, it is of no scientific or technical interest.
